# QUASI-POTENTIAL THEORY FOR ESCAPE PROBLEM: QUANTITATIVE SHARPNESS EFFECT ON SGD'S ESCAPE FROM LOCAL MINIMA

## ABSTRACT

We develop a quantitative theory on an escape problem of a stochastic gradient descent (SGD) algorithm and investigate the effect of sharpness of loss surfaces on the escape. Deep learning has achieved tremendous success in various domains, however, it has opened up various theoretical open questions. One of the typical questions is why an SGD can find parameters that generalize well over non-convex loss surfaces. An *escape problem* is an approach to answer this question, which investigates how efficiently an SGD escapes from local minima. In this paper, we introduce *quasi-potential* from traditional large deviation theory to the escape problem. Our novel formulation can quantify escaping effect without relying on auxiliary variables and is applicable to including discrete setup. Our theory find that (i) sharpness of a minimum exponentially slows down the escaping, (ii) but the SGD's noise cancels the effect, which leads to exponentially fast escape from sharp minima, as suggested by (Xie et al., 2020). We also conduct experiments to empirically validate our theory using neural networks trained with real data.

## 1 INTRODUCTION

In recent years, the successes of deep learning have been a major driving force of machine learning development (LeCun, 2019). Owing to its strong generalization capability, deep learning has diverged into a wide range of domains, such as computer vision (Krizhevsky et al., 2012), speech recognition (Mikolov et al., 2011), and natural language processing (Collobert et al., 2011). The high performance of deep learning is underpinned by gradient-based learning algorithms, including stochastic gradient descent (SGD) and its variations (Kingma & Ba, 2014; Schmidt et al., 2021). However, at the same time, those unprecedented successes raise a question:

*Why does SGD learn parameters of neural networks with high generalization performance?*

Although the optimization problems of neural networks were thought to be difficult to solve (Blum & Rivest, 1992), SGD can find nearly optimal solutions empirically, and further, the obtained solutions generalize well (Keskar et al., 2016; Brutzkus et al., 2017). Analyzing SGD's role on deep learning is an area of research that is currently attracting strong interest (Masters & Luschi, 2018; Jastrzebski et al., 2021).

One of the promising directions for this question is to study the geometric properties of a training loss landscape. Many empirical studies have found that minima obtained by SGD have distinctive geometric properties. Keskar et al. (2016) have shown that the shape of the minima obtained by SGD tends to be flat. He et al. (2019b) have deepened the investigation by picturing that SGD settles on the flatter side of asymmetric loss surface, which they named "asymmetric valley." Draxler et al. (2018) and Garipov et al. (2018) have shown that separate minima obtained by independent training processes are internally connected through pathways. Li et al. (2017) have proposed a dimension reduction technique to visualize the geometry of loss surfaces, visually confirming flat minima. Most significantly, Jiang et al. (2019) conducted large-scale experiments and verified that minima in flat and wide regions have the strongest correlations with generalization capabilities. To attain a theoretical understanding of SGD, it is key to quantitatively analyze the connection between SGD and the geometric properties of loss surfaces.

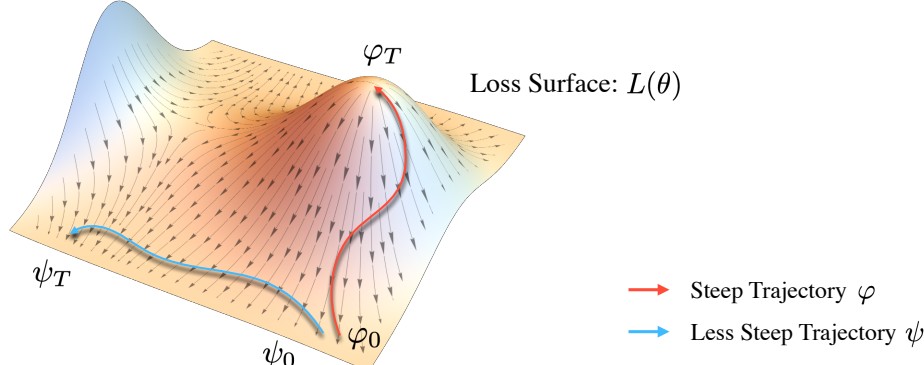

Figure 1: Visual illustration of steepness (Definition 3). The steepness of $\varphi$, $S_{0T}(\varphi)$, is greater than $S_{0T}(\psi)$ because $\varphi$ moves against the vector field of gradient $-\nabla L(\theta)$.

An *escape problem* is a scheme to analyze the dynamic of SGD escaping from local minima (Zhu et al., 2019; Jastrzębski et al., 2017; Hu et al., 2019; Nguyen et al., 2019; Xie et al., 2020). This scheme allows us to investigate why SGD avoids (potentially) bad local minima and settles on good minima. Zhu et al. (2019) first investigated the SGD's escape phenomenon and showed that SGD's escape is enhanced by its unique noise structure, called the "anisotropic noise structure." Invoked by their analysis, many studies have been attempting to theoretically quantify this phenomenon. Hu et al. (2019) rigorously identified the role of learning rate in escaping. Nguyen et al. (2019) used the Levy process to provide the precise description of SGD as well as its escaping phenomena. Jastrzębski et al. (2017) developed a theory of stochastic differential equation and quantified how the anisotropic noise affects its fast escape from sharp minima. Xie et al. (2020) refined the mathematical aspect and showed that the SGD's noise structure exponentially enhances escaping under a setup of diffusion theory.

In this paper, we introduce a *quasi-potential theory* to the escape problem, and investigate a *mean exit time*, which formally quantifies escaping. The notion of quasi-potential is defined in a fundamental theory of stochastic dynamical systems, named a Large Deviation Theory (Freidlin & Wentzell, 2012; Dembo & Zeitouni, 2010). Quasi-potential can formulate the distribution of trajectories that a stochastic dynamical system takes. To illustrate quasi-potential for SGD's escaping problem, we introduce an intuitive notion, *steepness* of a trajectory (Fig. 1 and Definition 3), and show that it is an effective tool to analyze the SGD's escaping. To the best of our knowledge, this is the first work that applies the quasi-potential to formalize the relationship between SGD's escape and sharpness.

Our main findings and contributions are as follows:

- We develop a novel quasi-potential theory that rigorously describes the escape of SGD with no auxiliary variables. Our theory can concisely incorporate the effect of essential factors, i.e., a batch size ($B$), a learning rate ($\eta$), and geometric parameters of loss surfaces ($r$ and $\lambda$)

- Our theory can be flexibly applied to several practical setups: SGD with discrete update and state-dependent gradient noise, while those were sometimes omitted in previous works for mathematical convenience.

- We find that a loss surface with sharp minima slows down the escape of SGD, which seems to contradict the common knowledge, i.e., SGD escapes efficiently from sharp minima. We show that our result does not contradict the common knowledge but is a generalization of existing results, such as (Jastrzębski et al., 2017) and (Xie et al., 2020).

## 1.1 COMPARISON WITH EXISTING STUDIES ON ESCAPE PROBLEM

In Table 1, we compare the escape time derived with the results of other studies that analyze the escape problem. There are two main points of focus. The first is the time to escape that we derive. Our theory provides a unified analysis of exit time incorporating all the essential parameters, batch

| Study | Time to escape | Non-stationary | Parameter dependent noise | Discrete setup |
|---|---|---|---|---|
| Hu et al. (2017) | $\propto \exp\left[\eta^{-1}\right]$ | $\checkmark$ | $\checkmark$ | |
| Jastrzębski et al. (2017) | $\exp\left[\frac{B}{2\eta}\Delta L\right]\sqrt{\frac{\det H'}{\det H}}$ | | $\checkmark$ | |
| Zhu et al. (2019) | N/A | $\checkmark$ | $\checkmark$ | |
| Nguyen et al. (2019) | $\frac{\alpha}{2}\frac{r^\alpha}{\sqrt{\eta}^\alpha}\left(1+\mathcal{O}\left(\eta^{\delta/2}\right)\right)$ | $\checkmark$ | | $\checkmark$ |
| Xie et al. (2020) | $\frac{2\pi}{|\lambda'|}\exp\left[2\frac{B}{\eta}\Delta L\left(\frac{s}{\lambda}+\frac{1-s}{|\lambda'|}\right)\right]$ | | $\checkmark$ | |
| **Ours** (continuous SGD) | $\exp\left[2\frac{B}{\eta}r^2\lambda^{\frac{1}{2}}\right]$ | $\checkmark$ | $\checkmark$ | |
| **Ours** (discrete SGD) | $\exp\left[2\frac{B}{\eta}r^2\lambda^{\frac{1}{2}}\right]+\mathcal{O}\left(\sqrt{\eta}\right)$ | $\checkmark$ | $\checkmark$ | $\checkmark$ |

Table 1: Comparison of the studies on the escape problem. $B$ is batch size, $\eta$ is a learning rate, $r$ is a radius of the region around a minimum, $H$ is a Hesse matrix of loss functions at a minima, and $\lambda = \lambda_{\min}(H)$. Further, $H'$ is a Hesse matrix on one of the neighboring points of the minimum and $\lambda'$ is one of its eigenvalues of $H'$. $\Delta L$ is a difference of training loss values within a neighborhood of minimum, $\alpha \in (0, 2]$ is an index of heavy-tailedness of gradient noise in SGD, and $\delta \in (0, 1)$ and $s \in (0, 1)$ are values that implicitly include various factors of the escaping problem. "Non-stationary" denotes whether the result holds without assuming that SGD reaches a stationary distribution before escaping. "Parameter dependent noise" denotes whether noise in SGD depends on current parameters. "Discrete setup" means whether the analysis is valid with a discrete update by SGD. Our theory has two main advantages: (i) it explicitly quantifies essential factors of escape without relying on auxiliary variables, such as $s, \delta$ and $\Delta L$, and (ii) it is applicable to a wide range of the settings. Finally, we note that although our results's dependency on $\lambda$ seems to be opposite to (Jastrzębski et al., 2017) and (Xie et al., 2020), those are all consisitent because $\Delta L$ has an implicit dependency on $\lambda$ and $r^2$ under Assumption 1.

size, learning rate, a radius of the region around a minimum, and sharpness of a minimum. As a consequence, we show that the eigenvalues of the Hesse matrix increase the time to escape, which has not been found in other studies. Such effect was less apparent in previous studies such as (Jastrzębski et al., 2017) and (Xie et al., 2020) because part of the sharpness dependency is hidden in $\Delta L$. Provided that $\Delta L = r^2\lambda$ under Assumption 1, one can see that our results are consistent with previous studies.

The second is our theory's flexibility. Different from Jastrzębski et al. (2017) and Xie et al. (2020), our theory does not require that SGD reaches the stationary distribution before escaping, which is known to take exponentially many iterations (Xu et al., 2017; Raginsky et al., 2017). Additionally, our theory can evaluate the correspondence with the practical SGD, which has a discrete update rule and state-dependent noise.

**Notations**: For a $k \times k$ matrix $A$, $\lambda_j(A)$ is the $j$-th largest eigenvalue of $A$, and $\lambda_{\max}(A) = \lambda_1(A)$ and $\lambda_{\min}(A) = \lambda_k(A)$ denote the largest and the smallest eigenvalue of a square matrix. $\mathcal{O}(\cdot)$ denotes Landau's Big-O notation. $\|\cdot\|$ denotes the Euclidean norm. Given a time-dependent function $\theta_t$, $\dot{\theta}_t$ denotes the differentiation of $\theta_t$ with respect to $t$. $N(\mu, \Sigma)$ denotes the multivariate Gaussian distribution with the mean $\mu$, and the covariance $\Sigma$.

## 2 SETTING AND PROBLEM

### 2.1 STOCHASTIC GRADIENT DESCENT AND DYNAMICAL SYSTEM

Consider a learning model parameterized by $\theta \in \mathbb{R}^d$. Given training examples $\{x_i\}_{i=1}^N$ and a loss function $\ell(\theta, x_i)$, we consider a training loss $L(\theta) := \frac{1}{N}\sum_{i=1}^N \ell(\theta, x_i)$ and a mini-batch loss

$L^B(\theta) := \frac{1}{B}\sum_{x_i \in \mathcal{B}} \ell(\theta, x_i)$, where $\mathcal{B}$ is a randomly sampled subset of the training data such that $|\mathcal{B}| = B$. We assume that $L(\theta)$ is differentiable and its derivative $\nabla L(\theta)$ is Lipschitz continuous.

We mainly consider two types of SGD: a discrete SGD and a continuous SGD. Although a discrete SGD is used in practice, we study continuous SGD as a starting point of our analysis because of its mathematical convenience. This is a widely used approach in general SGD analyses (Ali et al., 2019; Advani et al., 2020) as well as in the escaping analyses (Jastrzębski et al., 2017; Xie et al., 2020).

**Discrete SGD**: First, we give the usual discrete formulation of SGD. Given an initial parameter $\theta_0 \in \mathbb{R}^d$, SGD generates a sequence of parameters $\{\theta_k\}_{k \in \mathbb{N}}$ by the following update rule:

$$\theta_{k+1} = \theta_k - \eta \nabla L^B(\theta_k), \tag{1}$$

for $k \in \mathbb{N}$, where $\eta > 0$ is a learning rate.

In particular, we focus on SGD whose noise on gradients has a Gaussian distribution. We decompose $-\nabla L^B(\theta_k)$ in (1) into a gradient term $-\nabla L(\theta_k)$ and a noise term $\nabla L(\theta_k) - \nabla L^B(\theta_k)$, and consider a case that the noise is Gaussian. With this setting, the update rule in (1) is rewritten as

$$\theta_{k+1} = \theta_k - \eta \nabla L(\theta_k) + \sqrt{\frac{\eta}{B}} W_k, \tag{2}$$

where $W_k \sim N(0, \eta C(\theta_k))$ is a parameter-dependent Gaussian noise with its covariance $C(\theta) := \mathbb{E}_{i \sim \mathrm{Uni}(\{1,\dots,N\})} \left[ (\nabla L(\theta) - \nabla \ell(\theta, x_i))^\top (\nabla L(\theta) - \nabla \ell(\theta, x_i)) \right]$. We assume that $C(\theta)$ is Lipschitz continuous.

The Gaussianity of the noise on gradients is justified by the following reasons: (i) if the batch size $B$ is sufficiently large, the central limit theorem ensures the noise term becomes Gaussian noise, and (ii) several empirical studies show that the noise term becomes Gaussian noise (Mandt et al., 2016; Jastrzębski et al., 2017; He et al., 2019a), although different findings have been obtained in other settings (Simsekli et al., 2019).

**Continuous SGD**: We also give a continuous SGD, which is exactly discretized to (2) by a classic Euler scheme (Definition 5.1.1 of Gobet (2016)). With a time index $t \geq 0$ and the given initial parameter $\theta_0 \in \mathbb{R}^d$, the continuous dynamic of SGD is written as follows:

$$\dot{\theta}_t = -\nabla L(\theta_t) + \sqrt{\frac{\eta}{B}} C(\theta_t)^{1/2} w_t \tag{3}$$

where $w_t$ is a $d$-dimensional Wiener process, i.e. an $\mathbb{R}^d$-valued stochastic process with $t$ such that $w_0 = 0$ and $w_{t+u} - w_t \sim N(0, uI)$ for any $t, u > 0$. We note this system can be seen as a Gaussian perturbed dynamical system with a noise magnitude $\sqrt{\frac{\eta}{B}}$ because $\eta$ and $B$ do not evolve by time.

## 2.2 ESCAPE PROBLEM AND MEAN EXIT TIME

We consider the problem on how SGD escapes from minima of loss surfaces. In this paper, our target of interest is quantified by a notion of *mean exit time* for continuous SGD and *discrete mean exit time* of discrete SGD. Let $\theta^* \in \mathbb{R}^d$ be a local minimum of loss surfaces, and $D \subset \mathbb{R}^d$ be a $r$-neighborhood of $\theta^*$ with $r > 0$. We define the mean exit time as follows:

**Definition 1** (Mean exit time from $D$). *Consider a continuous SGD starting from $\theta_0 \in D$. Then, a mean exit time of the continuous SGD (3) from $D$ is defined as*

$$\mathbb{E}[\tau] := \mathbb{E}[\min\{t : \theta_t \notin D\}].$$

**Definition 2** (Discrete mean exit time from $D$). *Consider a discrete SGD starting from $\theta_0 \in D$. Then, a discrete mean exit time of the discrete SGD (2) from $D$ is defined as*

$$\mathbb{E}[\nu] := \mathbb{E}[\min\{k\eta : \theta_k \notin D\}].$$

These definitions are common in quasi-potential theory (Freidlin & Wentzell, 2012; Gobet, 2016). Intuitively, the smaller $\mathbb{E}[\tau]$ or $\mathbb{E}[\nu]$ becomes, the faster the system escapes from a region $D$. In other words, the system has a stronger tendency to escape from $\theta^*$.

We remark that there are other formulations to analyze the escape problem. Zhu et al. (2019) define escaping efficiency as $\mathbb{E}_{\theta_t}[L(\theta_t) - L(\theta_0)]$. Jastrzębski et al. (2017) and Xie et al. (2020) study a ratio between the probability of coming out from $\theta^*$'s neighborhood and the probability mass around $\theta^*$.

## 2.3 BASIC ASSUMPTIONS FOR THE ESCAPE PROBLEM

We provide basic assumptions for the escape problem, commonly used in the literature (Mandt et al., 2016; Zhu et al., 2019; Jastrzębski et al., 2017; Xie et al., 2020).

**Assumption 1** ($L(\theta)$ is locally quadratic). *There exists a matrix $H^* \in \mathbb{R}^{d \times d}$ such that for any $\theta \in D$, the following equality holds:*

$$\forall \theta \in D, L(\theta) = L(\theta^*) + \nabla L(\theta^*)(\theta - \theta^*) + \frac{1}{2}(\theta - \theta^*)^\top H^*(\theta - \theta^*)$$

**Assumption 2** (Hesse covariance matrix). *For any $\theta \in D$, $C(\theta)$ is approximately equal to $H^*$.*

It is known that Assumption 2 holds around a critical point $\theta^*$ (Zhu et al., 2019; Jastrzębski et al., 2017). It is also empirically shown that Assumption 2 can approximately hold even for randomly chosen $\theta$ (see Section 2 of Xie et al. (2020)). We further investigate a variant of Assumption 2 in Section 5.

## 3 QUASI-POTENTIAL THEORY

We introduce the basic notions of the quasi-potential theory. We start with defining a notion of *steepness* of a trajectory followed by the systems (3) on a loss surface $L(\theta)$. Let $\varphi = \{\varphi_t\}_{t \in [0,T]}$ be a trajectory over a finite time interval $[0, T]$. Or more formally, $\varphi$ is a continuous map from $[0, T]$ to $\mathbb{R}^d$, and is an element of $\mathbf{C}_{0T}(\mathbb{R}^d)$ which is a support of a dynamical process in $[0, T]$. Given a trajectory $\varphi$ and the system (3), we define the following quantity:

**Definition 3** (Steepness). *Steepness of a trajectory $\varphi$ followed by (3) is defined as*

$$S_{0T}(\varphi) := \frac{1}{2} \int_0^T (\dot{\varphi}_t + \nabla L(\varphi_t))^\top C(\varphi_t)^{-1/2} (\dot{\varphi}_t + \nabla L(\varphi_t)) \, dt.$$

Steepness $S_{0T}(\varphi)$ can be intuitively interpreted as the hardness of climbing that the system (3) is exposed to while following the trajectory $\varphi$ on $L(\theta)$ (Fig. 1). This notion is generally utilized in the field of dynamical systems, for example, and is called "normalized action functional" in Section 3.2 of Freidlin & Wentzell (2012) and "rate function" in Section 1.2 of Dembo & Zeitouni (2010).

Steepness is useful to describe a distribution of trajectories generated by dynamical systems. If a trajectory $\varphi$ has a large steepness $S_{0T}(\varphi)$, the probability that the dynamic system takes the trajectory decreases exponentially. Formally, the distribution is analyzed as follows.

**Lemma 1** (Theorem 3.1 in Section 3.3 Freidlin & Wentzell (2012)). *For any $\delta, \zeta > 0, \varphi \in \mathbf{C}_{0T}(\mathbb{R}^d)$, and sufficiently small $\varepsilon > 0$, the following holds:*

$$\mathrm{P}_{\varphi'}(\varphi' \in \mathbf{C}_{0T}(\mathbb{R}^d) \mid \rho(\varphi', \varphi) < \delta) \geq \exp\{-\varepsilon^{-2}[S_{0T}(\varphi) + \zeta]\},$$

*where $\rho(\varphi', \varphi) = \sup_{t \in [0,T]} \|\varphi'_t - \varphi_t\|$.*

**Lemma 2** (Theorem 3.1 in Section 3.3 Freidlin & Wentzell (2012)). *Let $\Phi(s) = \{\varphi \in \mathbf{C}_{0T}(\mathbb{R}^d) \mid S_{0T}(\varphi) \leq s\}$. For any $\delta, \zeta, s > 0$ and sufficiently small $\varepsilon > 0$, we have:*

$$\mathrm{P}_{\varphi'}\{\varphi' \in \mathbf{C}_{0T}(\mathbb{R}^d) \mid \rho(\varphi', \Phi(s)) \geq \delta\} \leq \exp\{-\varepsilon^{-2}(s - \zeta)\},$$

*where $\rho(\varphi', \Phi(s)) = \inf_{\varphi \in \Phi(s)} \rho(\varphi', \varphi)$.*

Although we restrict our attention to the system (3), the same discussion is applicable to a general class of diffusion processes and dynamical systems with Markov perturbations (For details, see section 5.7 in Dembo & Zeitouni (2010) or Section 6.5 in Freidlin & Wentzell (2012)).

Although there are several trajectories from $\theta^*$ to $\theta \in D$ with different steepness, a dominating factor for mean exit time is the smallest steepness among them, which is called *quasi-potential*:

**Definition 4** (Quasi-potential). *Quasi-potential of $\theta \in D$ is defined as*

$$V(\theta) := \inf_{T > 0} \inf_{\varphi:(\varphi_0, \varphi_T) = (\theta^*, \theta)} S_{0T}(\varphi).$$

Similar to steepness, quasi-potential can be seen as the minimum effort the system (3) needs to climb from $\theta^*$ up to $\theta$ on $L(\theta)$. (For more details, see Section 5.3 of Freidlin & Wentzell (2012)).

## 4 MEAN EXIT TIME ANALYSIS FOR SGD

### 4.1 ASSUMPTIONS

To analyze the mean exit time, the quasi-potential theory requires several assumptions regarding the *stability* of the system (3) at $\theta^*$.

**Assumption 3** ($\theta^*$ is asymptotically stable). *For any neighborhood $U$ that contains $\theta^*$, there exists a small neighborhood $V$ of $\theta^*$ such that gradient flow with any initial value $\theta_0 \in V$ does not leave $U$ for $t \geq 0$ and $\lim_{t \to \infty} \theta_t = \theta^*$.*

**Assumption 4** ($D$ is attracted to $\theta^*$). *$\forall \theta_0 \in D$, gradient flow with initial value $\theta_0$ converges to $\theta^*$ without leaving $D$ as $t \to \infty$.*

where "gradient flow" means a continuous gradient descent defined as $\dot{\theta}_t = -\nabla L(\theta_t)$.

*Stability* is a commonly used notion in dynamical systems (Hu et al., 2017; Wu et al., 2017), although it does not always appear in SGD's escaping analysis (Zhu et al., 2019; Jastrzębski et al., 2017; Xie et al., 2020). Assumption 3 is known to be equivalent to the local minimality of $\theta^*$ under the condition that $L(\theta)$ is real analytic around $\theta^*$ (Absil & Kurdyka, 2006). Also, by definition of asymptotic stability in Assumption 3, we can always find a region $D$ that satisfies Assumption 4. The more detailed properties of stability can be found, such as in Section 6.5 of Teschl (2000). Assumption 3 and 4 are necessary to obtain the result (5) in the following section.

Also, we require the following assumption as a boundary condition of Theorem 4.

**Assumption 5.** $L(\theta^*) = 0$

Assumption 5 is only for simplifying our proofs without changing the essence of our problem.

Finally, we assume the following,

**Assumption 6.** *For a potential field $W(\theta) : \mathbb{R}^d \mapsto \mathbb{R}$,*

$$\frac{1}{2}\nabla W(\theta)^\top C(\theta)^{1/2} \nabla W(\theta) - \nabla L(\theta)^\top \nabla W(\theta) = 0, W(\mathbf{0}) = 0$$

*has a unique solution.*

### 4.2 MAIN RESULTS

In preparation, we start with a traditional theorem in large deviation theory.

We analyze the mean escape time of SGD under the above assumptions. In preparation, we state two facts. First, as shown in Hu et al. (2019) Theorem 3.4, $V(\theta)$ is calculated as a solution of the following Hamilton-Jacobi equation,

$$\frac{1}{2}\nabla V(\theta)^\top C(\theta)^{1/2} \nabla V(\theta) - \nabla L(\theta)^\top \nabla V(\theta) = 0. \tag{4}$$

Second, if $\frac{\eta}{B}$ is sufficiently small, the mean exit time can be expressed using $V(\theta)$ as

$$\mathbb{E}[\tau] = \exp\left[\frac{B}{\eta}V_0\right], \tag{5}$$

where $V_0 := \min_{\theta' \in \partial D} V(\theta')$. Although these facts have been investigated in the literature (for example, see Section 4.4 in Freidlin & Wentzell (2012)), we give our own theorems and proofs in Appendix A and B for completeness.

The followings are our main results. We start with the mean exit time of Continuous SGD. Let $\mathbb{E}[\tau_{\mathrm{SGD}}]$ be the mean exit time of the continuous SGD, and let $\mathbb{E}[\tau_{\mathrm{isoSGD}}]$ be the mean exit time of an isotropic continuous SGD whose $C(\theta)$ is set to $I$.

**Theorem 1** (Continuous SGD). *Suppose that Assumption 1, 2, 3, 4, 5, and 6 hold. Then, for sufficiently small $\frac{\eta}{B}$,*

$$\mathbb{E}[\tau_{\mathrm{isoSGD}}] = \exp\left[2\frac{B}{\eta}r^2\lambda\right], \quad \mathbb{E}[\tau_{\mathrm{SGD}}] = \exp\left[2\frac{B}{\eta}r^2\lambda^{\frac{1}{2}}\right].$$

This result gives an exact expression for the expected escape time with the explicit values of SGD. The results also have two implications. First, these result both of those results show that the mean escape time exponentially increases in the smallest eigen value of $H^*$, i.e. $\lambda$. This implies that sharper minima generally slow down the escaping, which is seemingly opposite to the implication of the existing literature (Jastrzębski et al., 2017; Xie et al., 2020). But in fact this is consistent with the existing literature because some of the sharpness factor is implicitly included in other variables such as $\Delta L$. Second, our result endorses the fact that SGD's anisotropic noise exponentially accelerates the escaping (Xie et al., 2020), because the result shows that the mean exit time of SGD is smaller than that of isotropic SGD by $\exp[\lambda^{\frac{1}{2}}]$.

Our theory can be extended to the discrete case. By $\mathbb{E}[\nu_{\text{SGD}}]$, we denote the discrete mean exit time of the discrete SGD, and by $\mathbb{E}[\nu_{\text{isoSGD}}]$ we denote the one of an isotropic version of the discrete SGD, i.e. with $C(\theta)$ being $I$. The escaping problem of a discrete SGD, the discrete mean exit time, is formulated as a special case of Gobet & Menozzi (2010). By substituting $g(\cdot) = 0, f(\cdot) = 1$, and $k(\cdot) = 0$ in Theorem 17 in Gobet & Menozzi (2010), we can obtain the following simplified statement.

$$\max \left\{ \mathbb{E}[\nu_{\text{isoSGD}}] - \mathbb{E}[\tau_{\text{isoSGD}}], \mathbb{E}[\nu_{\text{SGD}}] - \mathbb{E}[\tau_{\text{SGD}}] \right\} = \mathcal{O}(\sqrt{\eta}).$$

which immediately prove the following theorem:

**Theorem 2** (Discrete SGD). *Given, Assumption 1, 2, 3, 4, 5, and 6, for sufficiently small $\frac{\eta}{B}$,*

$$\mathbb{E}[\nu_{\text{isoSGD}}] = \exp\left[2\frac{B}{\eta}r^2\lambda\right] + \mathcal{O}\left(\sqrt{\eta}\right), \quad \mathbb{E}[\nu_{\text{SGD}}] = \exp\left[2\frac{B}{\eta}r^2\lambda^{\frac{1}{2}}\right] + \mathcal{O}\left(\sqrt{\eta}\right)$$

This result suggests that the discrete error does not majorly affect the escape. We note this is the first study that confirms the validity of using a continuous SGD model (3) for escape analysis.

### 4.3 PROOF FOR THEOREM 1

We describe a proof of Theorem 3. We begin with the isotropic case and then investigate the non-isotropic case.

**Isotropic case** $\mathbb{E}[\tau_{\text{isoSGD}}]$**:** We substitute $I$ to $C(\theta)$. By the Jacobi equation (4) which is formally given by Theorem 4 in Appendix A, we have the following form for $\theta \in D$:

$$\frac{1}{2}\nabla V(\theta)^\top \nabla V(\theta) - \nabla L(\theta)^\top \nabla V(\theta) = 0. \tag{6}$$

Given $V(\theta^*) = 0$ by definition of $V(\theta)$, $V(\theta) = 2L(\theta)$ is the unique solution of (6) by Assumption 5 and 6. Therefore, we have

$$V_0 = \min_{x \in \partial D} 2L(\theta) = \min_{x \in \partial D} 2\theta^\top H^* \theta = 2r^2\lambda.$$

The second equality follows Assumption 1 and 5. Combined with the fact (5), which is formally shown in Theorem 5 in Appendix B, we obtain the statement of Theorem 1 in the isotropic case.

**Anisotropic case** $\mathbb{E}[\tau_{\text{SGD}}]$**:** Similar to the isotropic case, the equation (4) (or Theorem 4) gives

$$\frac{1}{2}\nabla V(\theta)^\top C(\theta)^{\frac{1}{2}}\nabla V(\theta) - \nabla L(\theta)^\top \nabla V(\theta) = 0. \tag{7}$$

$\nabla V(\theta) = 2C(\theta)^{-\frac{1}{2}}\nabla L(\theta)$ satisfies (7) and by Assumption 1 and 2 , $2C(\theta)^{-\frac{1}{2}}\nabla L(\theta)$ can simply written as

$$2C(\theta)^{-\frac{1}{2}}\nabla L(\theta) = 2H^{*-\frac{1}{2}}\nabla\left(\theta^\top H^*\theta\right) = 2H^{*-\frac{1}{2}}2H^*\theta = 4H^{*\frac{1}{2}}\theta.$$

Given $V(\theta^*) = 0$ by definition of $V(\theta)$, $V(\theta) = \theta^\top H^{*\frac{1}{2}}\theta$ is the unique solution of (7) by Assumption 5 and 6. Then, we obtain

$$V_0 = 2r^2\lambda_{\min}(H^{*\frac{1}{2}}), \tag{8}$$

Combined with (5), or Theorem 5 in Appendix B, we finish the proof of Theorem 1. $\qquad\square$

## 5 FURTHER INVESTIGATION ON COVARIANCE MATRIX

We investigate a variation of Assumption 2 on the covariance matrix. Although Assumption 2 is commonly used (Jastrzębski et al., 2017; Xie et al., 2020), relaxing this is important for a flexible modeling of SGD.

**Assumption 7** (Variant of Assumption 2). *$H^{*-1/2}$ is a positive matrix and there exist constants $0 < c_1 \leq c_2 < \infty$ such that for $\theta \in D$, $C(\theta) = H^*G$ holds with a positive invertible matrix $G$ as $0 < c_1 \leq \lambda_{\min}(G) \leq \lambda_{\max}(G) \leq c_2$.*

Under this different setup, we obtain the following results.

**Theorem 3.** *Suppose that Assumption 1, 3, 4, 5, 6, and 7 hold. Then, for sufficiently small $\frac{\eta}{B}$,*

$$\mathbb{E}[\tau_{\mathrm{isoSGD}}] = \exp\left[2\frac{B}{\eta}r^2\lambda\right]$$

$$\exp\left[2\frac{B}{\eta}r^2\frac{1}{\sqrt{c_2}}\lambda^{\frac{1}{2}}\right] \leq \mathbb{E}[\tau_{\mathrm{SGD}}] \leq \exp\left[2\frac{B}{\eta}r^2\frac{1}{\sqrt{c_1}}\lambda^{\frac{1}{2}}\right]$$

It suggests that replacing Assumption 2 by Assumption 7 does not affect the isotropic case, but has a constant effect in the anisotropic case.

*Proof for Theorem 3.* The result of $\mathbb{E}[\tau_{\mathrm{isoSGD}}]$ is obtained in the same way as Theorem 1. For the proof of $\mathbb{E}[\tau_{\mathrm{SGD}}]$, the following lemmas are useful, whose proofs are provided in Appendix C.

**Lemma 3.** *For positive invertible matrices $A$ and $B$, the following inequality holds,*

$$\lambda_{\min}(A^{-1}B^{-1}) \leq \lambda_{\max}(A^{-1})\lambda_{\min}(B^{-1})$$

**Lemma 4.** *For invertible matrices $A$ and $B$, the following inequality holds*

$$\lambda_{\min}(AB) \geq \lambda_{\min}(A)\lambda_{\min}(B)$$

Similar to the proof of Theorem 1, we obtain $\nabla V(\theta) = 2C(\theta)^{-\frac{1}{2}}\nabla L(\theta)$. Then, $\nabla V(\theta)$ is simply written as

$$\nabla V(\theta) = 2C(\theta)^{-\frac{1}{2}}\nabla L(\theta) = 2C(\theta)^{-\frac{1}{2}}\nabla\left(\theta^\top H^*\theta\right) = 2C(\theta)^{-\frac{1}{2}}2H^*\theta$$
$$= 2(H^*G)^{-\frac{1}{2}}2H^* = 4G^{-\frac{1}{2}}H^{*-\frac{1}{2}}H^*\theta = 4G^{-\frac{1}{2}}H^{*\frac{1}{2}}\theta.$$

The second equation follows Assumption 1 and 5, and the fourth equation follows Assumption 7. Given that $V(\theta^*) = 0$, we obtain $V(\theta) = \theta^\top G^{-\frac{1}{2}}H^{*\frac{1}{2}}\theta$ for $\theta \in D$. Then, we rewrite $V_0$ is as

$$V_0 = 2r^2\lambda_{\min}(G^{-\frac{1}{2}}H^{*\frac{1}{2}}), \tag{9}$$

By Lemma 3, we develop an upper bound on $\lambda_{\min}(G^{-1/2}H^{*1/2})$ as $\lambda_{\min}(G^{-1/2}H^{*1/2}) \leq \lambda_{\max}(G^{-1/2})\lambda_{\min}(H^{*1/2}) \leq c_1^{-1/2}\lambda^{1/2}$. Similarly, Lemma 4 gives us the following lower bound $\lambda_{\min}(G^{-1/2}H^{*1/2}) \geq \lambda_{\min}(G^{-1/2})\lambda_{\min}(H^{*1/2}) \geq c_2^{-1/2}\lambda^{1/2}$. We substitute the two inequalities into the solution (9), then obtain the following form of $V_0$:

$$2r^2\frac{1}{\sqrt{c_2}}\lambda^{\frac{1}{2}} \leq V_0 \leq 2r^2\frac{1}{\sqrt{c_1}}\lambda^{\frac{1}{2}}$$

Combined with (5), or Theorem 5 in Appendix B, we finish the proof of Theorem 3. □

## 6 EXPERIMENT

We conduct an experiment to validate our result of discrete setup (Theorem 2), using a neural network and real-world datasets. We use a multi-layer perception and the AVILA dataset (De Stefano et al., 2011) to observe that the discrete mean exit time of SGD has exponential dependence on eigenvalue $\lambda$, radius $r$ and a ratio of the learning rate and the batch size $\eta/B$.

In order for our essential assumptions to hold, we use the mean square loss with $\ell_2$ regularizer for $L(\theta)$ (Assumption 1) and train the model with the gradient descent for a sufficiently long time to

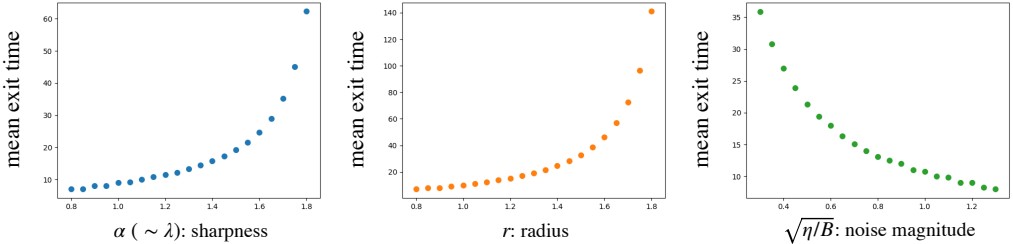

Figure 2: Empirical validation of Theorem 1, where the empirical mean exit time has exponential dependency on sharpness $\alpha$ ($\sim \lambda$), radius $r$, and noise magnitude $\sqrt{\eta/B}$.

obtain $\theta_0$ near $\theta^*$. We set the $r$-neighborhood of $\theta_0$ as $D$ (Assumption 3 and 4). To measure the discrete mean exit time, we repeatedly execute a vanilla SGD from $\theta_0$ for 1000 times and take an average number of steps at which SGD exit from $D$ (i.e. when the distance from $\theta_0$ becomes farther than $r$).

To control $\lambda$, we follow the approach of Xie et al. (2020). We obtain sharper minima by mapping the loss function $L$ to $L_\alpha$ such that $L_\alpha(\theta) := L(\sqrt{\alpha}\theta)$ ($\alpha > 0$) and setting $\theta_0 := \theta_0/\alpha$. Since this mapping coverts $\lambda$ to $\alpha\lambda$ with other properties remaining the same, we use $\alpha$ as a surrogate of $\lambda$.

We show the results in Figure 2. As Theorem 2 suggests, the noise magnitude $\sqrt{\eta/B}$ exponentially accelerates the escaping under our experiment setup, and eigen value and radius have the effect of exponentially slowing down the escaping

## 7 RELATED WORKS

We summarize relevant studies related to the topics on loss surfaces and the stochastic gradient descent algorithm. We mainly consider the following three factors.

**Loss surface shape**: Shape of loss surfaces have long been a topic of interest. The argument that the flatness of loss surfaces around local minima improves generalization was first studied by Hochreiter & Schmidhuber (1995; 1997), and the observation has recently reconfirmed in deep neural networks by Keskar et al. (2016). Sagun et al. (2017) empirically examined the flatness of loss surfaces. The theoretical advantage of the flatness was criticized by Dinh et al. (2017) in terms of scale-sensitivity of flatness, but Tsuzuku et al. (2020) and Rangamani et al. (2019) tackled the criticism by developing scale invariant flatness measures. An effect of the shape of loss surfaces on SGD was investigated in Wu et al. (2017); Ge et al. (2018), and Chaudhari et al. (2019); Foret et al. (2020) developed a variant of SGD which made use of this fact. In addition to the flatness, He et al. (2019b) proposed a new notion of asymmetry of loss surfaces, and Draxler et al. (2018); Garipov et al. (2018) studied how several local minima in a loss surface are connected. Li et al. (2018) developed a random dimensional reduction method to visualize loss surfaces on high dimensional spaces.

**Exit/Stability of SGD**: How SGD behaves in neighborhoods of local minima in loss surfaces is investigated from two aspects: stability and escape efficiency. For stability, a way in which SGD finds local minima and stabilizes was analyzed by Wu et al. (2018); Kleinberg et al. (2018); Achille et al. (2019); Li et al. (2017). Smith & Le (2017) used Bayesian ideas to analyze the stability. For exiting aspects, Jastrzębski et al. (2017) investigated an effect of a Hesse matrix of local minima on the ease of escaping ineffective local minima, and Xie et al. (2020) elaborated this effect via quantitative analysis. Zhu et al. (2019) showed that anisotropic structure of gradient noise by SGD is useful in escaping inefficient local minima, and Nguyen et al. (2019) studied an effect of non-Gaussianity of the gradient noise.

**SGD property**: Detailed nature of SGD itself is also an object of interest. The magnitude of the gradient noise by SGD is an important factor, including its relation to a learning rate and a batch size. An effect of large batch sizes on the reduction of gradient noise is investigated in Hoffer et al. (2017); Smith et al. (2018); Masters & Luschi (2018). Another area of interest is shape of a gradient noise distribution. Zhu et al. (2019); Hu et al. (2017); Daneshmand et al. (2018) investigated the anisotropic nature of gradient noise and its advantage. Simsekli et al. (2019) discussed the fact

that a gradient noise distribution has a heavier tail than Gaussian distributions. Nguyen et al. (2019); Şimşekli et al. (2019) showed benefits of these heavy tails for SGD. Panigrahi et al. (2019) rigorously examined gradient noise in deep learning and how close it is to a Gaussian. Xie et al. (2020) studied a situation where the distribution is Gaussian, and then analyzes the behavior of SGD in a theoretical way.

## 8 CONCLUSION

In this paper, we develop a novel quasi-potential theory for the escape problem of SGD. Our theory gives an intuitive picture of SGD's escaping dynamic, and also but is an effective tool for formal analysis. In our main result, our theory explicitly describes how the escape of SGD is affected by a batch-size, a learning rate, and radius of regions, and sharpness (Theorem 1). Furthermore, due to its flexibility, our theory allows the extended analyses, such as SGD's escape under even weaker assumption on covariance matrix (Theorem 3) and the escape problem of a discrete SGD (Theorem 2). We believe our theory provides a solid insight for SGD dynamics and also flexible theory for further studies.

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

## A    JACOBI EQUATION OF QUASI-POTENTIAL

**Theorem 4.** *For all $\theta \in D$, $V(\theta)$ satisfies the following Jacobi equation,*

$$\frac{1}{2}\nabla V(\theta)^\top C(\theta)^{1/2}\nabla V(\theta) - \nabla L(\theta)^\top \nabla V(\theta) = 0$$

*Proof of Theorem 4.* For $u, v \in \mathbb{R}^d$, we introduce an inner product and a norm regarding a point $\theta \in D$ as $\langle u, v\rangle_\theta := u^\top C(\theta)^{-1/2} v$ and $\|u\|_\theta := \sqrt{\langle u, u\rangle_\theta}$. With these definitions, the $S_{0T}(\varphi)$ is written as follows:

$$S_{0T}(\varphi) = \frac{1}{2}\int_0^T \|\dot{\varphi}_t + \nabla L(\varphi_t)\|_{\varphi_t}^2 \, dt. \tag{10}$$

Note that $\varphi_t \in D$ holds for any $t \in [0, T]$ by the definition of trajectories. We rewrite the integrand of (10) as follows:

$$\begin{aligned}
&\|\dot{\varphi}_t + \nabla L(\varphi_t)\|_{\varphi_t}^2 \\
&= \|\dot{\varphi}_t\|_{\varphi_t}^2 + \|\nabla L(\varphi_t)\|_{\varphi_t}^2 + 2\langle\dot{\varphi}_t, \nabla L(\varphi_t)\rangle_{\varphi_t} \\
&= \left(\|\dot{\varphi}_t\|_{\varphi_t} - \|\nabla L(\varphi_t)\|_{\varphi_t}\right)^2 + 2\|\dot{\varphi}_t\|_{\varphi_t}\|\nabla L(\varphi_t)\|_{\varphi_t} + 2\langle\dot{\varphi}_t, \nabla L(\varphi_t)\rangle_{\varphi_t} \\
&\geq 2\|\dot{\varphi}_t\|_{\varphi_t}\|\nabla L(\varphi_t)\|_{\varphi_t} + 2\langle\dot{\varphi}_t, \nabla L(\varphi_t)\rangle_{\varphi_t}. \tag{11}
\end{aligned}$$

We develop a parameterization for the term in (11). For a trajectory $\varphi$, we select an bijective function $f : [0, 1] \to [0, T]$ as satisfying the follows: for each $t \in [0, T]$ and $t^* \in [0, 1]$ as $t = f(t^*)$, a parameterized trajectory $\varphi_{t^*}^* := \varphi_{f(t^*)}$ satisfies

$$\|\dot{\varphi}_{t^*}^*\|_{\varphi_{t^*}^*} = \|\nabla L(\dot{\varphi}_{t^*}^*)\|_{\varphi_{t^*}^*}. \tag{12}$$

This parameterization reduces the quasi-potential to the minimum of the following quantity:

$$(S_{0T}(\varphi) \geq S_{0f(T)}(\varphi^*) =) \int_0^{f(T)} \|\dot{\varphi}_{t^*}^*\|_{\varphi_{t^*}^*}\|\nabla L(\varphi_{t^*}^*)\|_{\varphi_{t^*}^*} + \langle\dot{\varphi}_{t^*}^*, \nabla L(\varphi_{t^*}^*)\rangle_{\varphi_{t^*}^*} \, dt^* \tag{13}$$

subject to the constraint (12). Since the integrand of (13) includes the first order derivative regarding $t^*$, (13) holds over different parameterizations $f$. For convenience, we use another bijective parameterization function $g : [0, R] \to [0, 1]$ as $t^* = g(r)$ with $R > 0$ and $r \in [0, R]$ such that $\hat{\varphi}_r := \varphi^*_{g(r)}$ satisfies

$$\|\dot{\hat{\varphi}}_r\|_{\hat{\varphi}_r} = 1. \tag{14}$$

Then, the quasi-potential is reduced to the following formula, [1]

$$V(\theta) = \inf_{r \in [0,R]: \|\dot{\hat{\varphi}}_r\|_{\hat{\varphi}_r}=1} \int_0^R \left( \|\dot{\hat{\varphi}}_r\|_{\hat{\varphi}_r} \|\nabla L(\hat{\varphi}_r)\|_{\hat{\varphi}_r} + \left\langle \dot{\hat{\varphi}}_r, \nabla L(\hat{\varphi}_r) \right\rangle_{\hat{\varphi}_r} \right) dr, \tag{15}$$

where $\hat{\varphi}_R = \theta$. By the Bellman equation-type optimality, we expand the right hand side of (15) into the following form:

$$V(\theta) = \inf_{r \in [0,R]: \|\dot{\hat{\varphi}}_r\|_{\hat{\varphi}_r}=1} \left( \int_{R-\delta}^R \left( \|\dot{\hat{\varphi}}_r\|_{\hat{\varphi}_r} \|\nabla L(\hat{\varphi}_r)\|_{\hat{\varphi}_r} + \left\langle \dot{\hat{\varphi}}_r, \nabla L(\hat{\varphi}_r) \right\rangle_{\hat{\varphi}_r} \right) dr + V(\hat{\varphi}_{R-\delta}) \right), \tag{16}$$

with a width value $\delta > 0$. The Taylor expansion around $r = R$ gives

$$\int_{R-\delta}^R \left( \|\dot{\hat{\varphi}}_r\|_{\hat{\varphi}_r} \|\nabla L(\hat{\varphi}_r)\|_{\hat{\varphi}_r} + \left\langle \dot{\hat{\varphi}}_r, \nabla L(\hat{\varphi}_r) \right\rangle_{\hat{\varphi}_r} \right) dr + V(\hat{\varphi}_{R-\delta})$$

$$= \delta \left( \|\nabla L(\hat{\varphi}_R)\|_{\hat{\varphi}_R} + \left\langle \nabla L(\hat{\varphi}_R), \dot{\hat{\varphi}}_R \right\rangle_{\hat{\varphi}_R} - \dot{\hat{\varphi}}_R^\top \nabla V(\hat{\varphi}_R) \right) + V(\hat{\varphi}_R) + O(\delta^2)$$

Taking $\delta \to 0$ and noticing $\hat{\varphi}_R = \theta$, (16) can be simplified to the following equation:

$$0 = \inf_{r \in [0,R]: \|\dot{\hat{\varphi}}_r\|_{\hat{\varphi}_r}=1} \left( \|\nabla L(\theta)\|_\theta + \left\langle \nabla L(x)^\top, \dot{\hat{\varphi}}_R \right\rangle_\theta - \dot{\hat{\varphi}}_R \nabla V(\theta) \right) \tag{17}$$

It remains to select $\hat{\varphi}$ which solves the minimization problem (17). Since the following equality holds,

$$\left\langle \nabla L(\theta)^\top, \dot{\hat{\varphi}}_R \right\rangle_\theta - \dot{\hat{\varphi}}_R \nabla V(\theta) = \left\langle \nabla L(\theta)^\top - \nabla V(\theta)^\top C(\theta)^{-1/2}, \dot{\hat{\varphi}}_R \right\rangle_\theta, \tag{18}$$

it is easy to see that a trajectory $\hat{\varphi}^*$ such that

$$\dot{\hat{\varphi}}_R^* = -\frac{\nabla L(x)^\top - \nabla V(\theta) C(\theta)^{1/2}}{\|\nabla L(\theta)^\top - \nabla V(\theta) C(\theta)^{1/2}\|_\theta}$$

minimizes (18). With this $\hat{\varphi}^*$, (17) simplifies to

$$\|\nabla L(\theta)\|_\theta = \|\nabla L(\theta)^\top - \nabla V(\theta) C(\theta)^{1/2}\|_\theta.$$

Taking the square of both sides, we get the statement. $\qquad\square$

## B  THEOREM ON EXIT TIME

In this section, we develop the following fundamental theorem on the notion of exit time. For simplicity, we use $\varepsilon$ to denote $\sqrt{\eta/B}$.

**Theorem 5.** *If $\varepsilon$ is sufficiently small,*

$$\mathbb{E}[\tau] = \exp\left[\varepsilon^{-2} V_0\right]$$

*holds, where $V_0 := \min_{\theta' \in \partial D} V(\theta')$*

---

[1]One might think that if we parametrize as above (14), the equality condition for (11) is violated. Indeed

$$S_{0T}(\hat{\varphi}) \neq \int_0^T \|\dot{\varphi}_t\|_{\hat{\varphi}_t} \|\nabla L(\hat{\varphi}_t)\|_{\hat{\varphi}_t} + \langle \dot{\varphi}_t, \nabla L(\hat{\varphi}_t) \rangle_{\hat{\varphi}_t} dt$$

for $\hat{\varphi}$. However, $\hat{\varphi}$ is introduced just for the simple calculation of $S_{0T}(\varphi^*)$. Although $\hat{\varphi}$ frequently appears in the proof, our attention is still on $\varphi^*$ and $S_{0T}(\varphi^*)$, not on $S_{0T}(\hat{\varphi})$.

To prove this result, we provide the proof for an upper bound (Lemma 6) and a lower bound (Lemma 7). Throughout the proofs, we use $\mathbf{C}_{0T,\theta_0}$, $\mathrm{P}_{\theta_0}$, instead of $\mathbf{C}_{0T}$ or $\mathrm{P}$, to clearly indicate which trajectory we are referring to.

We introduce several notions. For $\delta > 0$ and $\theta \in \mathbb{R}^d$, let $\mathcal{B}_\delta(\theta)$ denote an $\delta$-neighbourhood of $\theta$, that is, $\mathcal{B}_\delta(\theta) := \{\theta' \in \mathbb{R}^d \mid \|\theta' - \theta\| \leq \delta\}$. Further, for a set $\Theta \subset \mathbb{R}^d$, $\mathcal{B}_\delta(\Theta) := \cup_{\theta \in \Theta} \mathcal{B}_\delta(\theta)$.

The following lemma provides preliminary facts for proofs.

**Lemma 5.** *For any $c > 0$, there exist $\mu_1, \mu_2, T_1, T_2 > 0$ such that the followings hold:*

1. $\forall \theta \in D$, *there exists a trajectory $\varphi^1$ such that $\varphi_0^1 = \theta$, $\varphi_T^1 \in \mathcal{B}_{\mu_1/2}(\theta^*)$, $0 < T \leq T_1$ and $S_{0T}(\varphi^1) = 0$.*

2. $\forall \theta \in \mathcal{B}_{\mu_1}(\theta^*)$, *there exists a trajectory $\varphi^2$ such that $\varphi_0^2 = \theta$, $\varphi_T^2 \in \partial\mathcal{B}_{\mu_2}(D)$, $0 < T \leq T_2$ and $S_{0T}(\varphi^2) < V_0 + \frac{c}{2}$.*

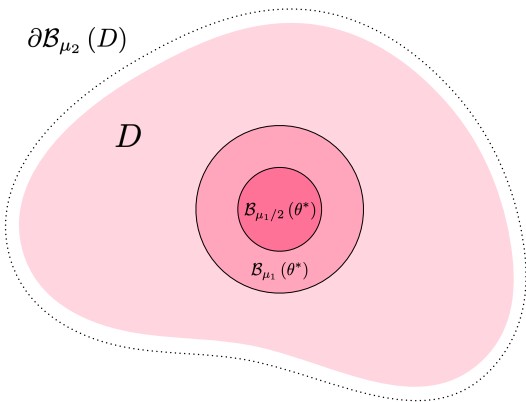

Figure 3: Illustration of domains and boundary, $D$, $\mathcal{B}_{\mu_1/2}(\theta^*)$, $\mathcal{B}_{\mu_1}(\theta^*)$, and $\partial\mathcal{B}_{\mu_2}(D)$

The illustration can be found in Fig. 3.

*Proof of Lemma 5.* The first statement immediately holds by the fact that $D$ is attracted to a asymptotically stable equilibrium position $\theta^*$ (Assumption 3 and 4).

For the second statement, since $V_0 := \min_{\theta' \in \partial D} V(\theta')$, there exists a trajectory $\varphi^a$ such that $\varphi_0^a = \theta^*$, $\varphi_{T_a}^a \in \partial D$ and $S_{0T_a}(\varphi^a) = V_0$, where $T_a$ is finite by Lemma 2.2 (a) in Freidlin & Wentzell (2012). We cut off the first portion of $\varphi^a$ up until the first intersecting point with $\mathcal{B}_{\mu_1}(\theta^*)$ and define it as $\varphi^b$. This means $\varphi_0^b \in \mathcal{B}_{\mu_1}(\theta^*)$, $\varphi_{T_b}^b \in \partial D$ and $S_{0T_b}(\varphi^b) < V_0$ hold. By Lemma 2.3 in Freidlin & Wentzell (2012), there exists a trajectory from $\varphi_{T_b}^b$ to a point $\theta_{\mu_2}$ in $\partial\mathcal{B}_{\mu_2}(D)$ such that the steepness is less than $K|\theta_{\mu_2} - \varphi_{T_b}^b|$ with a constant $K > 0$. Then, if we take a small enough $\mu_2$, we can obtain $\varphi^c$ such $\varphi_0^c = \varphi_{T_b}^b$, $\varphi_{T_c}^c \in \partial\mathcal{B}_{\mu_2}(D)$ and $S_{0T_c}(\varphi^c) < \frac{c}{2}$. By connecting $\varphi^b$ and $\varphi^c$, we obtain an appropriate $\varphi^2$. $\square$

**Lemma 6.** *If $\varepsilon > 0$ is sufficiently small,*

$$\mathbb{E}[\tau] = O\Big(\exp\big[\varepsilon^{-2}V_0\big]\Big)$$

*holds, where $V_0 := \min_{\theta' \in \partial D} V(\theta')$.*

*Proof of Lemma 6.* We show that for any constant $c > 0$, there exists a small $\varepsilon_0$ such that $\forall \varepsilon \leq \varepsilon_0$ such that $\mathbb{E}[\tau] \leq \exp\big[\varepsilon^{-2}(V_0 + c)\big]$. To the aim, we split the dynamical system (3) of our interest into the first half and the second half, $\{\theta_t^1\}_t$ and $\{\theta_t^2\}_t$. $\{\theta_t^1\}_t$ starts with $\theta_0^1 = \theta_0 \in D$ and terminates when it first reaches $\mathcal{B}_{\mu_1/2}(\theta^*)$. We define the terminating time of $\{\theta_t^1\}_t$ as $\tau_1 := \min\{t > 0 : \theta_t^1 \in \mathcal{B}_{\mu_1/2}(\theta^*)\}$. On the other hand, $\{\theta_t^2\}_t$ starts with $\theta_0^2 = \theta_{\tau_1} \in \mathcal{B}_{\mu_1/2}(\theta^*)$ and terminates when it first

reaches $\partial D$. We define the terminating time of $\{\theta_t^2\}_t$ as $\tau_2 := \min\{t > 0 : \theta_t^1 \in \partial D\}$. Clearly, the exit time $\tau = \tau_1 + \tau_2$.

Regarding $\tau_1$ and $\tau_2$, we show the following two independent facts with sufficiently small $\varepsilon > 0$.

Fact 1 : $\tau_1$ is no more than $T_1$ with probability at least $1/2$.

Fact 2 : $\tau_2$ is no more than $T_2$ with probability at least $\exp\left\{-\varepsilon^{-2}\left(V_0 + c\right)\right\}$.

**Fact 1:** Given the trajectory $\varphi_1$ provided by Lemma 5, Lemma 1 gives us that if $\varepsilon < \varepsilon_{\text{stp1}}(\varphi_1, \mu_1/2, 1)$, the following inequality holds

$$P_{\theta_0^1}\left\{\varphi' \in \mathbf{C}_{0T_1, \theta_0^1}(\mathbb{R}^d) \mid \rho\left(\varphi', \varphi_1\right) < \mu_1/2\right\} \geq \exp\left\{-\varepsilon^{-2}\right\}.$$

Therefore, if we take $\varepsilon < \min\{\sqrt{1/\ln 2}, \varepsilon_{\text{stp1}}(\varphi_1, \mu_1/2, 1)\}$, we have

$$P_{\theta_0^1}\left\{\varphi' \in \mathbf{C}_{0T_1, \theta_0^1}(\mathbb{R}^d) \mid \rho\left(\varphi', \varphi_1\right) < \mu_1/2\right\} \geq \frac{1}{2}$$

Since the event of $\{\varphi' \in \mathbf{C}_{0T_1, \theta_0^1}(\mathbb{R}^d) \mid \rho\left(\varphi', \varphi_1\right) < \mu_1/2\}$ means that $\{\theta_t^1\}_t$ reaches $\mathcal{B}_{\mu_1}(\theta^*)$ in no later than $T_1$, we obtain the following which provides Fact 1.

$$P_{\theta_0^1}\left\{\tau_1 < T_1\right\} \geq \frac{1}{2}. \tag{19}$$

**Fact 2:** Given the trajectory $\varphi_2$ provided by Lemma 5, Lemma 1 tells us that if $\varepsilon < \varepsilon_{\text{stp1}}(\varphi_2, \mu_2, c/2)$, the following inequality holds

$$P_{\theta_0^2}\left\{\varphi' \in \mathbf{C}_{0T_2, \theta_0^2}(\mathbb{R}^d) \mid \rho\left(\varphi', \varphi_2\right) < \mu_2\right\} \geq \exp\left\{-\varepsilon^{-2}\left(S_{0T_2}\left(\varphi_2\right) + \frac{c}{2}\right)\right\}.$$

$\{\varphi' \in \mathbf{C}_{0T_2, \theta_0^2}(\mathbb{R}^d) \mid \rho\left(\varphi', \varphi_2\right) < \mu_2\}$ is the event that $\{\theta_t^2\}_t$ goes out of $D$ in no more than the time $T_2$. Also, we know that $S_{0T_2}\left(\varphi_2\right) < V_0 + \frac{c}{2}$ by Lemma 5. Hence, we can conclude the following for Fact 2:

$$P_{\theta_0^2}\left\{\tau_2 < T_2\right\} \geq \exp\left\{-\varepsilon^{-2}\left(V_0 + c\right)\right\}. \tag{20}$$

Combining (19) and (20), we can obtain

$$P_{\theta_0}\left\{\tau < T_1 + T_2\right\} \geq \frac{1}{2}\exp\left\{-\varepsilon^{-2}\left(V_0 + c\right)\right\}. \tag{21}$$

Since this is a simple exponential distribution, we can obtain the following expectation:

$$\mathbb{E}\left[\tau\right] \leq 2\left(T_1 + T_2\right)\exp\left\{\varepsilon^{-2}\left(V_0 + c\right)\right\}$$

By setting

$$\varepsilon < \frac{1}{\sqrt{\ln 2(T_1 + T_2)}}\min\left\{\sqrt{\frac{1}{\ln 2}}, \varepsilon_{\text{stp1}}(\varphi_1, \mu_1/2, 1), \varepsilon_{\text{stp1}}(\varphi_2, \mu_2, c/2)\right\},$$

we can get

$$\mathbb{E}\left[\tau\right] \leq \exp\left\{\varepsilon^{-2}\left(V_0 + c\right)\right\}.$$

Then, we obtain the statement. $\qquad\square$

Next, we develop the lower bound on the exit time.

**Lemma 7.** *If $\varepsilon > 0$ is sufficiently small,*

$$\mathbb{E}\left[\tau\right] = \Omega\left(\exp\left[\varepsilon^{-2}V_0\right]\right)$$

*holds, where $V_0 := \min_{\theta' \in \partial D} V(\theta')$.*

*Proof of Lemma 7.* We show for any positive constant $c > 0$, there exists a small $\varepsilon_0$ such that $\forall \varepsilon \leq \varepsilon_0, \mathbb{E}[\tau] \geq \exp\left[\varepsilon^{-2}(V_0 - c)\right]$ holds.

We consider a specific case where the initial value of (3) is in $\partial \mathcal{B}_{\mu_1/2}(\theta^*)$, which can be trivially extended to general cases. Consider a Markov chain $Z_k$ ($k \in \mathbb{N}$) as a discretization of $\theta_t$ as $t = \tau_k$ with a $k$-th time grid $\tau_k$. It is formally defined as follows:

1. $\tau_0 = 0$,

2. $\sigma_k = \inf\{t > \tau_k \ : \ \theta_t \in \partial \mathcal{B}_{\mu_1}(\theta^*)\}$,

3. $\tau_k = \inf\left\{t > \sigma_{k-1} : \theta_t \in \partial \mathcal{B}_{\mu_1/2}(\theta^*) \cup \partial D\right\}$,

4. $Z_k = \theta_{\tau_k}$.

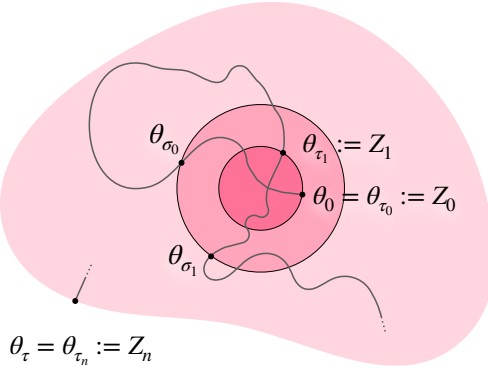

Figure 4: A continuous trajectory $\theta_t$ and the Markov chain $Z_k$ generated from $\{\theta_t\}_t$. Colored domains indicate $D$, $\mathcal{B}_{\mu_1/2}(\theta^*)$, and $\mathcal{B}_{\mu_1}(\theta^*)$ as illustrated in Fig. 3.

By introducing $Z_k$, we can reduce the continuous process $\{\theta_t\}_t$ to a discrete Markov chain transiting between $\partial \mathcal{B}_{\mu_1/2}(\theta^*)$ and $\partial D$. The illustration can be found in Fig. 4.

Let $\kappa := \inf\{k \mid Z_k \in \partial D\}$. Then, we have $\tau = \tau_n$ and

$$\mathbb{E}[\tau] = \sum_{k=0}^{\infty} \left( P_{\theta_0}\{\kappa \geq k\} - P_{\theta_0}\{\kappa \geq k+1\} \right)\tau_k = \sum_{k=1}^{\infty} P_{\theta_0}\{\kappa \geq n\}(\tau_k - \tau_{k-1}).$$

This can be further evaluated as

$$\mathbb{E}[\tau] > \sum_{k=1}^{\infty} P_{\theta_0}\{\kappa \geq k\}(\tau_k - \sigma_{k-1}) > \sum_{k=1}^{\infty} P_{\theta_0}\{\kappa \geq n\} \left( \inf_{\theta_0 \in \partial \mathcal{B}_{\mu_1}(\theta^*)} \mathbb{E}[\tau_1] \right),$$

which follows $\tau_{k-1} < \sigma_{k-1}$. Since $\mathcal{B}_{\mu_1/2}(\theta^*)$ is a strict subset of $\mathcal{B}_{\mu_1}(\theta^*)$, and $\mathcal{B}_{\mu_1}(\theta^*)$ is a strict subset of $D$, it takes a positive amount of time to transit from $\partial \mathcal{B}_{\mu_1}(\theta^*)$ to either $\partial \mathcal{B}_{\mu_1/2}(\theta^*)$ or $\partial D$, and there exists a positive lower bound $t_1$ for $\inf_{\theta_0 \in \mathcal{B}_{\mu_1}(\theta^*)} \mathbb{E}[\tau_1]$ that is independent of $\varepsilon$. Thus we get

$$\mathbb{E}[\tau] > t_1 \sum_{k=1}^{\infty} P_{\theta_0}\{\kappa \geq k\}.$$

By Lemma 8, we immediately get $P_{\theta_0}\{\kappa > k\} \geq [1 - \exp\{-\varepsilon^{-2}(V_0 - c)\}]^{k-1}$, hence we have

$$\mathbb{E}[\tau] > t_1 \sum_{k=1}^{\infty} \left[1 - \exp\left\{-\varepsilon^{-2}(V_0 - c)\right\}\right]^{k-1} = t_1 \exp\left\{\varepsilon^{-2}(V_0 - c)\right\}.$$

This implies $\mathbb{E}[\tau] \geq \exp\{\varepsilon^{-2}(V_0 - c)\}$ holds if $\varepsilon$ is small enough. □

**Lemma 8.** *We obtain*

$$\mathrm{P}(Z_{k+1} \in \partial D \mid Z_k \in \mathcal{B}_{\mu_1/2}(\theta^*)) \leq \exp\left\{-\varepsilon^{-2}(V_0 - c)\right\}.$$

*Proof of Lemma 8.* First, we decompose $\mathrm{P}(Z_{k+1} \in \partial D \mid Z_k \in \partial\mathcal{B}_{\mu_1/2}(\theta^*))$ into two parts in the following way:

$$
\begin{aligned}
&\mathrm{P}_{\theta_0}(Z_{k+1} \in \partial D \mid Z_k \in \partial\mathcal{B}_{\mu_1/2}(\theta^*)) \\
&\leq \max_{\theta_0' \in \partial\mathcal{B}_{\mu_1/2}(\theta^*)} P_{\theta_0'}\{\tau_1 = \tau\} \\
&= \max_{\theta_0' \in \partial\mathcal{B}_{\mu_1/2}(\theta^*)} \left[ \mathrm{P}_{\theta_0'}\{\tau = \tau_1 < T\} + \mathrm{P}_{\theta_0'}\{\tau = \tau_1 \geq T\} \right]
\end{aligned}
\tag{22}
$$

This holds for arbitrary $T$, so we pick $T = T'$ large enough so that this inequality holds for the first term:

$$\mathrm{P}_{\theta_0'}\{\tau = \tau_1 \geq T'\} \leq \frac{1}{2}\exp\left\{-\varepsilon^{-2}(V_0 - c)\right\} \tag{23}$$

The existence of such $T'$ is guaranteed by the fact that $V_0$ is finite and the following lemma.

**Lemma 9** (Lemma 2.2 (b) in Freidlin & Wentzell (2012)). *For any $\alpha > 0$, there exists positive constants $c$ and $T_0$, such that for all sufficiently small $\varepsilon > 0$ and any $\theta_0 \in D \cup \partial D \backslash \mathcal{B}_\alpha(\theta^*)$ we have the inequality*

$$\mathrm{P}_{\theta_0}\{\zeta_\alpha > T\} \leq \exp\left\{-\varepsilon^{-2}c(T - T_0)\right\},$$

*where $\zeta_\alpha = \inf\{t : \theta_t \notin D\backslash\mathcal{B}_\alpha(\theta^*)\}$.*

Given a constant $T'$, we consider bounding $\mathrm{P}_{\theta_0'}\{\tau = \tau_1 < T'\}$. Consider the following set of trajectories:

$$\Phi(V_0 - c/2) := \{\varphi : \varphi_0 = \theta_0', S_{0T}(\varphi) \leq V_0 - c/2\}.$$

Since it takes at least $V_0$ to reach $\partial D$ from $\theta^*$, the following inequality holds:

$$\mathrm{P}_{\theta_0'}\{\tau = \tau_1 < T'\} \leq \mathrm{P}_{\theta_0'}\{\varphi_y \notin \Phi(V_0 - c/2)\}.$$

Also, Lemma 2 implies, for all $\varepsilon \leq \varepsilon_{\mathrm{stp2}}(V_0 - c/2, \delta, c/2)$

$$
\begin{aligned}
\mathrm{P}_{\varphi'}\left\{\varphi' \in \mathbf{C}_{0T,\theta_0'}(\mathbb{R}^d) \mid \rho\big((\varphi' - \Phi(V_0 - c/2)) \geq \delta\big)\right\} &\leq \exp\{-\varepsilon^{-2}((V_0 - c/2) - c/2)\} \\
&= \exp\{-\varepsilon^{-2}(V_0 - c)\}
\end{aligned}
$$

Since $\delta$ can be arbitrarily small, the event of $\{\varphi_y \notin \Phi(V_0 - c/2)\}$ is equal to the event of $\{\varphi' \in \mathbf{C}_{0T,\theta_0'}(\mathbb{R}^d) \mid \rho((\varphi' - \Phi(V_0 - c/2)) \geq \delta\}$. Hence, we obtain

$$\mathrm{P}_{\theta_0'}\{\tau = \tau_1 < T\} < \exp\{-\varepsilon^{-2}(V_0 - c)\}. \tag{24}$$

If we set $\varepsilon \leq \frac{1}{\sqrt{\ln 2}}\varepsilon_{\mathrm{stp2}}(V_0 - c/2, \delta, c/2)$, we conclude

$$\mathrm{P}_{\theta_0'}\{\tau = \tau_1 < T\} < \frac{1}{2}\exp\{-\varepsilon^{-2}(V_0 - c)\}. \tag{25}$$

Combining (22), (23), and (25), we prove the statement. □

## C  PROOFS FOR LEMMA 3 AND LEMMA 4

*Proof of Lemma 3.* We introduce several definitions only for this proof. Supposing that we have a vector $x = (x_1, \ldots, x_n) \in \mathbb{R}^n$, $x^\downarrow$ and $x^\uparrow$ denote vectors whose coordinates are obtained by rearranging the numbers $x_j$ in decreasing order and in increasing order respectively, that is, $x^\downarrow = (x_1^\downarrow, \ldots, x_n^\downarrow)$ and $x^\uparrow = (x_1^\uparrow, \ldots, x_n^\uparrow)$, where $x_1^\downarrow \geq \cdots \geq x_n^\downarrow$ and $x_1^\uparrow \leq \cdots \leq x_n^\uparrow$. Also, given $x, y \in \mathbb{R}^n$, we denote $x \prec y$, if

$$\sum_{j=1}^n x_j^\downarrow = \sum_{j=1}^n y_j^\downarrow, \text{ and } \sum_{j=1}^k x_j^\downarrow \leq \sum_{j=1}^k y_j^\downarrow \text{ for } 1 \leq \forall k \leq n.$$

For an $n \times n$ matrix $A$, $\lambda(A) = (\lambda_1(A), \lambda_2(A), ..., \lambda_n(A))$. Also, we define its elementwise logarithm $\log \lambda(A) = (\log \lambda_1(A), \log \lambda_2(A), ..., \log \lambda_n(A))$.

Corollary III.4.6 in (Bhatia, 1997) claims that given two positive matrices $A$ and $B$, we have

$$\log \lambda^{\uparrow}(A) + \log \lambda^{\downarrow}(B) \prec \log \lambda(AB) \prec \log \lambda^{\downarrow}(A) + \log \lambda^{\downarrow}(B)$$

Following the definition of $\prec$, the left part, $\log \lambda^{\uparrow}(A) + \log \lambda^{\downarrow}(B) \prec \log \lambda(AB)$, can be equivalently restated as

$$\sum_{j=1}^{k} \left\{ \log \lambda^{\uparrow}(A) + \log \lambda^{\downarrow}(B) \right\}_j^{\downarrow} \leq \sum_{j=1}^{k} \left\{ \log \lambda(AB) \right\}_j^{\downarrow} \text{ for } 1 \leq \forall k \leq n$$

For the case $k = 1$, we have

$$\left\{ \log \lambda^{\uparrow}(A) + \log \lambda^{\downarrow}(B) \right\}_1^{\downarrow} \leq \left\{ \log \lambda(AB) \right\}_1^{\downarrow}$$
$$\Longleftrightarrow \max_{1 \leq i \leq n} \left\{ \log \lambda_{n-i+1}(A) + \log \lambda_i(B) \right\} \leq \log \lambda_{\max}(AB)$$

Since $\max_{1 \leq i \leq n} \left\{ \log \lambda_{n-i+1}(A) + \log \lambda_i(B) \right\} \geq \log \lambda_n(A) + \log \lambda_1(B) = \log \lambda_{\min}(A) + \log \lambda_{\max}(B)$ holds, we have $\log \lambda_{\min}(A) + \log \lambda_{\max}(B) \leq \log \lambda_{\max}(AB)$, or equivalently

$$\lambda_{\min}(A)\lambda_{\max}(B) \leq \lambda_{\max}(AB).$$

Since $\lambda_{\max}(A) = 1/\lambda_{\min}(A^{-1})$ , we have

$$\frac{1}{\lambda_{\max}(A^{-1})} \frac{1}{\lambda_{\min}(B^{-1})} \leq \frac{1}{\lambda_{\min}((AB)^{-1})}$$
$$\Longleftrightarrow \lambda_{\min}((AB)^{-1}) \leq \lambda_{\max}(A^{-1})\lambda_{\min}(B^{-1}).$$

Since commuted matrices share eigenvalues, $\lambda_{\min}((AB)^{-1}) = \lambda_{\min}(B^{-1}A^{-1}) = \lambda_{\min}(A^{-1}B^{-1})$, we have

$$\lambda_{\min}(A^{-1}B^{-1}) \leq \lambda_{\max}(A^{-1})\lambda_{\min}(B^{-1}).$$

$\square$

*Proof of Lemma 4.* $\lambda_{\max}(A)$ is equal to the spectral norm of $A$. By the sub-multiplicative property of spectral norm, we obtain

$$\lambda_{\max}(A)\lambda_{\max}(B) \geq \lambda_{\max}(AB),$$

Therefore, since $\lambda_{\max}(A) = 1/\lambda_{\min}(A^{-1})$,

$$\lambda_{\max}(A^{-1})\lambda_{\max}(B^{-1}) \geq \lambda_{\max}(A^{-1}B^{-1})$$
$$\Longleftrightarrow \lambda_{\max}(A^{-1})\lambda_{\max}(B^{-1}) \geq \lambda_{\max}((AB)^{-1})$$
$$\Longleftrightarrow \frac{1}{\lambda_{\min}(A)} \frac{1}{\lambda_{\min}(B)} \geq \frac{1}{\lambda_{\min}(AB)}$$
$$\Longleftrightarrow \lambda_{\min}(AB) \geq \lambda_{\min}(A)\lambda_{\min}(B).$$

Then, we obtain the statement. $\square$

