# OpenReview forum: "Quasi-potential theory for escape problem: Quantitative sharpness effect on SGD's escape from local minima"
_ICLR.cc/2022/Conference — ICLR 2022 Submitted_

### Official Review · Reviewer_NNov · 2021-10-31

**Correctness:** 1
**Technical Novelty And Significance:** 1
**Empirical Novelty And Significance:** Not applicable
**Recommendation:** 3
**Confidence:** 5

**Main Review:**

There is a deep inconsistency while using the quasi-potential theory to study the escape properties of such dynamical processes as SGD. The point is that quasi-potential, or in general the Freidlin-Wentzell type large deviations theory, handles "large scale" or "global" deviations from equilibrium positions. However, in the computable cases the authors must make assumptions on the local properties of the loss landscape, such as their Assumption 1, which essentially says that the loss function is quadratic. Such quadratic approximations can only hold near the equilibrium point (by Morse theory). Thus at a large scale, explicit analytical calculations based on quasi-potential for quadratic loss landscape is not applicable to the escape from local minima of SGD, since such escape is about the "far from equilibrium" situation. There is also an inadequacy of citing references, since very similar approaches and even the same examples of quadratic loss landscape have already been considered in Hu et al (2019). The major reason why the authors of Hu et al (2019) did not formally publish their work, but remains it as a preprint, is somehow mainly due to the same inconsistency issue pointed out here.

**Summary Of The Paper:**

This work proposes to use the notion of quasi-potential to analyze the dynamics and escape properties of SGD from local minima. Authors expect to use the quasi-potential theory to obtain a quantitative criteria for the ability of escape from local minima.

**Summary Of The Review:**

There is a deep inconsistency in the theory proposed in this paper.

---

> ### Author Response · Authors · 2021-11-22
> **Thank you for your review. We have addressed your concern.**
>
> Thank you for scrutinizing our paper. We have addressed your comments below.
>
> Q1. Hu et al. (2019) is not cited adequately. A similar analysis with a similar example can be found.
>
> A. Thank you for pointing it out. In our new manuscript, we have clearly specified that we used Theorem 3.4 of Hu et al. (2019) for our analysis. However, we'd also like to emphasize that our work has a contextual difference from Hu et al. (2019). While Hu et al. (2019)'s focus is general quantification of the SGD's implicit regularization, our central focus is the relationship between sharpness and SGD's escape. Given that sharpness is an active research target in deep learning theory [Jiang et al. 2019], our work has a substantial contribution, answering how sharpness affects SGD's escaping.
>
> About the similar example (Example 3.1. in [Hu et al. 2019]), we note that quadratic loss landscape is used for our own necessity. In our work, quadratic loss landscapes inevitably appear because we focus on the second-order derivative of loss surface, i.e., sharpness. On the other hand, Example 3.1. in [Hu et al. 2019] was used for an illustration.
>
> Q2. The quasi-potential analysis is not applicable to locally quadratic minima
>
> A. Although our analysis is not applicable to the SGD's global dynamics, our results are consistent around a minimum, which gives a solid clue for a famous research question, "why can SGD reach flat minima?" Our answer is "because SGD's noise enhances escape from sharp minima." We also note that Jastrzebski et al. (2017) and Xie et al. (2020) also limit their analyses to the local dynamics to answer the question above.
>
> Reference
>
> 1. Yiding Jiang, Behnam Neyshabur, Hossein Mobahi, Dilip Krishnan, and Samy Bengio. Fantastic generalization measures and where to find them. arXiv preprint arXiv:1912.02178, 2019.
> 2. Wenqing Hu, Zhanxing Zhu, Haoyi Xiong, and Jun Huan. Quasi-potential as an implicit regularizer for the loss function in the stochastic gradient descent. arXiv preprint arXiv:1901.06054, 2019.
> 3. Xie, Zeke, Issei Sato, and Masashi Sugiyama. "A Diffusion Theory For Deep Learning Dynamics: Stochastic Gradient Descent Exponentially Favors Flat Minima." International Conference on Learning Representations. 2020.
> 4. Jastrzebski, Stanislaw, et al. "Catastrophic fisher explosion: Early phase fisher matrix impacts generalization." International Conference on Machine Learning. PMLR, 2021.

---

### Official Review · Reviewer_5Jw7 · 2021-11-01

**Correctness:** 3
**Technical Novelty And Significance:** 3
**Empirical Novelty And Significance:** 2
**Recommendation:** 3
**Confidence:** 4

**Main Review:**

I am not absolutely familiar with the literature of escape time theory for SGD. And, first, I would like to state two important points about this approach:
 - it is not clear for me how it really relates to the explainability of good generalization properties of SGD as stated in the introduction on page 1 in italic center-text. I would have liked more motivation on this fact in the main text. Maybe with some concrete examples that such phenomena exist in practice (escaping bad local minima very fast for example).
- it is also not clear for me how Assumptions 1 and 2 are good models for what happens in practice. Especially for Assumption 2, for which it has been importantly remarked in S.Wojtowytsch [ https://arxiv.org/abs/2106.02588 ] and Pesme et al. [ https://arxiv.org/abs/2106.09524 ] that the covariance is degenerate near optima and hence cannot be equal to $H^*$ (except if one adds some random noise intentionally in the data set or if one considers the infinite data limit -which is not explained in the article).

I truly understand that these are share problems with already published literature, but still, I am not very convince about these results for the reasons above.

Leaving this discussion on the side, let us now comment a bit more the precise results of the paper under review.

First, I have to say that the paper is not very clearly written: for exemple important paragraphs such as the end of the abstract or the exposition of the main contributions are
 fuzzy. It makes the reader confused as it is difficult to know *what results* are going to be proven, *what* is the new contribution of the article, and *why* it is interesting. An archetypal example of this fact is that the authors claim to have developed a "novel quasi-potential theory that rigorously describes the escape of SGD": in fact, quasi potential theory is a *theoretical methodology* that helps in exhibiting escape rates, whereas it is often presented as a contribution in itself in the article (even if obviously large deviations is an important theory that should help understanding some of the behavior of SGD).

Second, it is very troubling that Table 1 offers comparison with some similar studies without explaining precisely the difference in the assumption and/or the results. More importantly, as results of the present article go sometimes in opposite directions with the bounds of Table 1, it is very weird to have almost zero comments on it. Is the bound of the present article in contradiction with the other ones ? stated in a much restricted setting ? only valid in some limit ? For this reason, comparison is, under the current form, almost impossible to make.

Thirdly, and maybe more importantly, the content appears to be mathematically not very precise if not sometimes incorrect: a mild example of this fact is that theorems are stated as equality whereas is seems obvious that they are in fact asymptotic first order developments with respect to parameter $\eta/B$ at the exponential scale (as it is always the case in large deviation theory). Another example is that the well known Hamilton-Jacobi equation Eq (4), is not clearly introduced: no name, no references, no mention of the fact that there is under general assumption no unicity...

Finally, the statements of the theorems are not enough put in perspective with practice and precise SGD dynamics behavior. Do we really see this dynamical phenomenon *during the training of SGD* (and not, as the experiments show, if initialized near a minimum) ?





**Summary Of The Paper:**

The paper under review gives mean exit time rates in the small step-size limit of SGD under some quadratic approximation of the loss. The analysis rests on standard results of Freindlin -Wentzell large deviation theory.

**Summary Of The Review:**

To conclude, the results of the presented paper may be of great interest, but under the current form, without any proper mathematical formulation, more rigorous comparison, or further perspectives with SGD behavior, I consider that this paper should be revised before resubmission.

---

> ### Author Response · Authors · 2021-11-22
> **Thank you for your thorough feedback. Indeed, our paper does have a flaw.**
>
> Thank you very much for your thorough review. We realized our paper's notation was incorrect (Q1). As you suggested, we would like to revise and resubmit in the future.
>
> We, nonetheless, have answered the other questions and revised our draft (Q2~8), which we thought we can address independently of the flaw. We would appreciate further discussion with you.
>
> Q1. Why are main theorems (Theorem 1, 2, and 3) stated as equality? Aren't those supposed to be the asymptotic results?
>
> A. We regard this as a critical mistake. We will work on revising them for future resubmission.
>
> Q2. Is the escape phenomenon observable in the practical SGD dynamics?
>
> A. The practical validity of the escaping phenomenon is still open to debate. We support its validity because of (i) the fact that the training loss of SGD usually shows fluctuation during training (e.g. Fig. 4 in [Huang et al., 2017) and (ii) that the SGD tends to move toward flatter minima under the practical setup (e.g. Fig. 3 left column in [Foret et al., 2020] and 2nd and 4th plots from the left of Fig. 4 in [Jastrzebski 2021]).
>
> Q3. Isn't Assumption 2 too strong?
>
> A. We agree that Assumption 2 is a strong assumption. But as you mentioned, it is a common assumption in existing works, and at least there exist ways to make it hold, such as assuming the infinite width of networks. We believe that our framework has a substantial theoretical interest.
>
> Q4. Abstract and Introduction are not very well written. It is hard to see what contribution this paper has.
>
> A. Thank you for the feedback. We have updated our abstract and introduction to highlight our contributions. We believe the itemized contributions in p2 would clarify for readers what to focus on.
>
> Q5. Your presentation sounds as if quasi-potential analysis itself is your contribution.
>
> A. We appreciate this feedback. We have fixed the misleading parts, but we still believe that Section 3 has substantial contributions. We intended to provide an intuitive introduction of quasi-potential and its effectiveness. Given that quasi-potential is not a well-known notion in the machine learning community, our paper would work to initiate a solid understanding of quasi-potential.
>
> Q6. Your results seem to be contradicting previous works.
>
> A. Essentially those are not contradicting. It is true that our results and Jastrzebski et al. (2017) and Xie et al.(2020) appear to have the opposite dependency on $\lambda$. But the apparent inconsistency comes from an auxiliary variable $\Delta L$ (a difference of training loss values within a neighborhood of minimum). Since $\Delta L$ has an implicit dependency on $\lambda$ and $r^2$ under Assumption 1, our main result can be written as follows too, which is consistent with Jastrzebski et al. (2017) and Xie et al.(2020).
> $$\exp \left[2 \frac{B}{\eta} \Delta L \lambda^{-\frac{1}{2}}\right]$$
>  We have updated the caption to avoid confusion.
>
> Q7. Hamilton-Jacobi equation Eq (4), is not clearly introduced.
>
> A. We agree that we should have described the theoretical background of Ep. (4) in more detail. We have clarified that this is the Hamilton-Jacobi equation before Ep. (4).
>
> Q8. For Eq. (4), there is no mention of the fact that there is under the general assumption no unicity.
>
> A. Thank you for pointing this out. We have added the additional assumption that Eq. (4) has a unique solution (Assumption 6).
>
> Reference
>
> 1. Huang, Gao, et al. "Densely connected convolutional networks." Proceedings of the IEEE conference on computer vision and pattern recognition. 2017.
> 2. Foret, Pierre, et al. "Sharpness-aware Minimization for Efficiently Improving Generalization." International Conference on Learning Representations. 2020.
> 3. Xie, Zeke, Issei Sato, and Masashi Sugiyama. "A Diffusion Theory For Deep Learning Dynamics: Stochastic Gradient Descent Exponentially Favors Flat Minima." International Conference on Learning Representations. 2020.
> 4. Jastrzebski, Stanislaw, et al. "Catastrophic fisher explosion: Early phase fisher matrix impacts generalization." International Conference on Machine Learning. PMLR, 2021.

---

### Official Review · Reviewer_mKRS · 2021-11-02

**Correctness:** 4
**Technical Novelty And Significance:** 3
**Empirical Novelty And Significance:** Not applicable
**Recommendation:** 5
**Confidence:** 3

**Main Review:**

1- The paper is well written and easy to follow. The experimental results are confirming the theoretical results.

2- The assumptions 2,3 are still strong and unlike the Xi et al 2020 paper, the covariance C(\theta) depends on the Hess of minimizer and is independent of \theta.

3- In Xi et. al 2020 the exit time depends on the inverse of sharpness but in your results, it depends on it directly. How can one explain this discrepancy?

4- The analysis just considers the SGD with fixed step size however in practice we use decreasing step size to train DNNs. What are the challenges of analyzing this scenario?

5- The exit time depends exponentially on the sharpness. Now, if at the minimum \lamba=0 and we have wide minima then based on your results the trajectory should jump out of that region instantly which contradicts our observation in practice. How can we explain this based on your theoretical results?


**Summary Of The Paper:**

This paper uses the quasi potential theory to formalize the escape behavior of SGD happening while training a deep neural network. The quasi potential is defined based on the steepness of a trajectory and is the smallest steepness to go from the minimizer to another point. Utilizing this they analyze both continuous and discrete SGD and show that the average exit time for a trajectory of SGD to move out of the neighborhood of minima depends exponentially on the mini-batch size, the sharpness of the minima, the radius of the neighborhood, and the inverse of learning rate.

**Summary Of The Review:**

The analysis just considers the SGD with fixed step size however in practice we use decreasing step size to train DNNs. What are the challenges of analyzing this scenario?

---

> ### Author Response · Authors · 2021-11-22
> **Thank you for your thorough review. We have addressed your comment of decreasing step size as well as others.**
>
> Thank you for your thorough review. We have addressed your questions below.
>
> Q1. Isn't Assumption 2 stronger than the one in Xie et al. 2020?
>
> A. In fact, Xie et al. (2020) use a similar assumption implicitly. In Appendix A.2 of Xie et al., they say "The temperature $T_a$ dominates the path $a \to s$ (p16)", where "$T_{a}=\frac{\eta}{2 B} H_{a}$ (p17)". That means the Hessian $H_a$ (as well as $T_a$) is constant along an escaping trajectory. Combined with this implicit assumption, their assumption on $C(\theta)$ is fairly similar to our Assumption 2.
>
>
> Q2. What are the challenges of analyzing in the scenario of decreasing step size?
>
> A. Thank you for the comment. We used a constant step size because our analysis focuses on the local dynamics of SGD, where step size does not significantly decrease. Please also refer to our answer for Q2 from Reviewer NNov for why we only consider local dynamics.
>
>  However, this is a theoretically interesting question. We suspect the theoretical challenge would be the following. If we try to incorporate the adaptive step size, $\eta$ in Eq. (3) becomes time-dependent $\eta(t)$. As a consequence, Eq. (3) becomes no longer a diffusion process, which is incompatible with quasi-potential theory (Theorem 5). Although there are exit time analyses for the general Markov processes (e.g. Section 6.5 in [Freidlin and Wentzell 2012]), it would be challenging to derive an enlightening result from it.
>
> Q3. Doesn't this result imply that smaller sharpness leads to faster escape?
>
> A. Indeed, our result implies small sharpness leads thus faster escape and this implication might look inconsistent with the common knowledge. However, this is a superficial discrepancy due to the difference in setup.
> We characterize the neighborhood of a minimum by the radius from a minimum. On the other hand, some of the previous works, such as Jastrzebski et al. (2017) and Xie et al.(2020), characterize the neighborhood by the height of the loss surface from a minimum. We adopted our setup so that it aligns with the setup of the quasipotential theory. (Our answer for Q6 from Reviewer 5Jw7 provides a quantitative explanation for this.)
>
> We also note that, regardless of the different setups, we and previous works both argue that "SGD's anisotropic noise enhances escape more than the isotropic noise does"
>
> Reference
> 1. Xie, Zeke, Issei Sato, and Masashi Sugiyama. "A Diffusion Theory For Deep Learning Dynamics: Stochastic Gradient Descent Exponentially Favors Flat Minima." International Conference on Learning Representations. 2020.
> 2. Jastrzebski, Stanislaw, et al. "Catastrophic fisher explosion: Early phase fisher matrix impacts generalization." International Conference on Machine Learning. PMLR, 2021.
> 3. Freidlin, Mark I., and Alexander D. Wentzell. 2012. Random Perturbations of Dynamical Systems. Berlin, Heidelberg: Springer Science & Business Media.

---

### Author Response · Authors · 2021-11-23
**Our Rebuttal is for Future Resubmission**

We thank the reviewers for their constructive comments and useful insights.
Thanks to those comments, we realize our work includes a critical issue (Q1 by Reviewer 5Jw7), which, we thought, is hard to fix during this submission.

In our rebuttal and revision, we have only addressed other comments that are independent of the aforementioned issue.
We appreciate further discussion for our future resubmission.

---

### Decision · Program_Chairs · 2022-01-20

**Decision:**

Reject

**Comment:**

The paper uses quasi-potential theory to analyze the escape behavior of SGD. Although this is a topic of interest to the ML community, the reviewers found a critical issue with the paper, which the authors admit can not be fixed during this submission. I, therefore, do not think there is a need for a longer discussion.